# Parallels between Postpartum Disorders in Humans and Preweaning Piglet Mortality in Sows

**DOI:** 10.3390/ani8020022

**Published:** 2018-02-06

**Authors:** Courtney Daigle

**Affiliations:** Department of Animal Science, Texas A&M University, College Station, TX 77843, USA; cdaigle@tamu.edu

**Keywords:** parturition, sow, human, postpartum depression, social network, piglet crushing

## Abstract

**Simple Summary:**

Humans and sows are both highly social species that exhibit a wide variety of maternal behaviors and responsivity to pregnancy and parturition. Piglet crushing is a production and welfare concern for the swine industry. Similar to rates of postpartum depression in humans, the performance of piglet crushing is more likely in first-time mothers. Furthermore, hormonal profiles and social factors that influence the development of this disease in humans mirror those observed in sows surrounding parturition. This article reviews the biological, social, and management factors that may be contributing to this problem of piglet crushing through the lens of how postpartum depression develops in humans. Utilizing knowledge from human psychology and animal welfare science may provide producers with management tools to mitigate piglet crushing and provide new insight into the factors that contribute to human postpartum disorders.

**Abstract:**

Pregnancy and parturition in all mammals is accompanied with physical, psychological, social, and hormonal shifts that impact the mother physically and psychologically. Pre-weaning piglet mortality continues to be a major welfare and economic issue in U.S. swine production, running at 12–15% with crushing by the sow the major cause. Much research has focused on farrowing environment design, yet the fact that little progress has been made emphasizes that psychosocial factors may impact rates of postpartum disorders (PPD). There is a mismatch between evolved adaptations and contemporary psychosocial and management practices. Many factors associated with the development of PPD in humans are mirrored in sows that perform piglet crushing. These factors include poor mental welfare (anxiety, difficulty coping with stress), a lack of experience, a lack of social support, and individual differences in their sensitivity to hormone concentrations. Understanding what strategies are effective in preventing PPD in humans may have welfare and production benefits for sows—and sows may be a possible model for better understanding PPD in humans.

## 1. Introduction

Pregnancy and parturition are experiences accompanied with large hormonal shifts that impact the physical and mental state of the mother, and as a result, some mothers may develop postpartum disorders (PPD), including postpartum depression and postpartum psychosis. Piglet savaging has been suggested to resemble postpartum psychosis [1], and if both crushing and savaging are related to affective problems, then there may be more long-term consequences for maternal behavior and future piglet development. Approximately 12% of sows will crush piglets within 72 h post-parturition, and primiparous sows (gilts) are more likely to savage their piglets compared to sows who are farrowing their second or third litter of piglets [2]. Although piglet crushing can be a single instance, or related to poor leg condition, poor piglet viability, or poor pen design, clearly, there are some sows who display aberrant maternal behavior, crush several piglets, and do not respond to screaming piglets during crushing [3,4]. Therefore, the performance of piglet savaging and piglet crushing may be different behavioral manifestations of postpartum disorders in sows.

Postpartum depression is unique compared to other types of depression in that it is preceded and accompanied by major biological adaptations that may also affect mood, and that it cannot only affect the mother, but can also impact the offspring’s cognitive, behavioral, and emotional development [5,6]—effects that can potentially last into adolescence [7]. Postpartum depression in humans, as defined by the American Psychiatric Association [8], is a major depressive disorder that includes any time during the pregnancy or the first four weeks postpartum. This disorder affects up to 15% of new mothers [9], is more common in first-time mothers [10], is heritable [11], is not accompanied by any characteristic precursors during pregnancy, and rates of recurrence can reach up to 50% [12].

Theoretical perspectives from evolutionary science offer explanations for PPD as a psychological adaptation or a byproduct of modern civilizations. Maternal infanticide is a counter-intuitive, counter-evolutionary behavior accompanied by serious welfare and economic ramifications. Hagen [13] proposed that PPD facilitated maternal disinvestment in offspring that are unlikely to survive and later reproduce, and that the development of PPD broadcasts the mother’s need for support. Predictors of PPD in humans are poor infant and maternal health and lack of social support. Recently, the current high rates of PPD in humans could be argued to be a byproduct of major changes in motherhood, where PPD may be a “disease of civilization” [14]. Some of the factors associated with the development of PPD include early weaning [15], low omega-3 fatty acid consumption [16], Vitamin D deficiency [17], sedentary lifestyles [18], and social isolation [19] are more prevalent today than in the past—and some of these changes in ‘civilization’ mirror the challenges sows face in commercial pig production. Examining the parallels in social network experiences, parity contributions, hormone shifts, and behavioral changes or restrictions between PPD in humans and piglet crushing in sows presents an opportunity to enhance our understanding of both issues, use lessons learned from humans to mitigate piglet crushing in sows, and perhaps increase our understanding of human PPD based upon what we can learn from sows.

## 2. Social Network

Larger social networks are more protective against PPD in humans [19,20,21,22]. Pregnancy and parturition, particularly for the primiparous mother, can be a difficult and jarring experience. Members of the animal kingdom that evolve social systems have the capacity to learn socially transmitted information (e.g., information that is learned by observing social conspecifics), have the capacity to communicate among members of the group, and are capable of experiencing a variety of affective and mental states, including depression [23]. Social support, such as perceived assistance from a partner and emotional closeness with other mothers in the first few weeks after parturition were observed to be key factors in predicting the development of PPD in mothers that were evaluated eight weeks postpartum [24]. Therefore, providing new human mothers with a strong social network of support is important to preventing the development of postpartum psychosis.

A similar argument can be made for pigs. Pigs are a social, highly cognitive species, and with the evolution of social structures comes the capacity to communicate with conspecifics and transfer social information either by experience or observation. Pigs will separate themselves from the group close to parturition, they may begin parturition well before delivery. Even though pigs will seek isolation the few days surrounding parturition, the social support prior to and after the sow rejoins the group may be invaluable to the success of the sow and her litter [25]. Free ranging pigs have been observed to organize themselves into small social clusters, or sounders, that usually are comprised of a number of adult sows with their daughters, and any unweaned young [26]. The nature of this social organization structure depicts a scenario in which young breeding-age females learn appropriate maternal behaviors from their mothers. Sows in natural conditions will leave the sounder at least 24 h prior to farrowing and will begin to construct a nest. Approximately 9–10 days post-farrowing, the sow will return to their sounder with the newborn piglets [27]. Group housed sows displayed improved maternal behavior when housed in lactation groups 3 days post-farrowing compared to sows housed in farrowing crates [25]. The development of housing and management technologies that allow for social contact surrounding farrowing [28] further emphasizes the need to better understand the social dynamics surrounding farrowing on sow and piglet welfare. While more research is needed evidence suggests that the social environment, physical contact opportunities, and information transfer may be valuable to the success and mitigation of PPD in the gilt. 

Therefore, since social isolation increases likelihood of PPD in humans [29], separating gilts from multiparous sows during gestation and at parturition may not be in the best interest of the sow. Since humans and pigs are both highly social, cognitive species, sows may require more social support during pregnancy and parturition than is currently available, and being housed singly without the opportunity to interact with other sows before, during, or immediately after farrowing may be counterproductive in preventing the occurrence of piglet crushing and savaging.

## 3. Impact of Parity

Both gilts and humans are at an increased risk of developing PPD. Primiparous humans are more likely to develop puerpareal psychosis [30], yet, the development of PPD in humans is multifactorial and likely epigenetic in development [31]. Gilts have been observed to savage more [32], and be more likely to crush their piglets compared to second and third-parity sows [33]. Human mothers with prior experience caring for infants are less likely to develop PPD [34], and rates of PPD in sows are lower in multiparous parity sows. However, consideration should be made to the impact of management practices because sows that crush a large number of piglets may be culled after a single parity.

For sows, the factors contributing to this phenomenon are multifactorial and are potentially exacerbated by the fact that gilts in commercial production systems may not have exposure to multiparous sows surrounding parturition, and may therefore, have no information about what to expect. Gilts may not have seen a piglet since they *were* a piglet, so the sudden appearance of a piglet after experiencing farrowing may be very stressful. The stressors associated with the novelty of parturition and movement from a familiar gestation environment to a novel farrowing environment, the discomfort associated with pain, the lack of understanding why they are experiencing the pain associated with farrowing, the impact of large hormonal swings, and the lack of social support in a highly social species may contribute to the development of a poor affective state. Therefore, lack of experience engaging with and caring for young may be a contributing factor to the manifestation of PPD in humans and sows.

## 4. Reproductive Hormones

### 4.1. Steroid Hormones

Steroid hormones may play a large role in the development of PPD. During human pregnancy, estriol increases 1000 fold [35], estradiol increases 50-fold, progesterone increases 10-fold [36], and prolactin increases 7-fold [37]. In sows, estrogen level swings can be as much as a 26-fold decrease from 2 days pre-farrow to 4 days post-farrow [38]. Both humans and sows have higher levels of estrogen products during gestation, and these hormone levels will drop drastically postpartum. Therefore, the impact of these hormone level changes may be similar in humans and sows.

In humans, these vast increases in hormone levels will return to pre-pregnancy levels within one to two weeks postpartum [37], and this sudden withdrawal of pregnancy hormones has been observed to stimulate depressive-like symptoms [39]. Further, the sudden withdrawal of estrogen, estrogen fluctuations, and sustained estrogen deficiencies have been associated with mood disturbances [40]. A similar phenomenon was observed in mice where a rapid withdrawal from estradiol and progesterone was associated with increases in anxiety and aggressive behaviors as well as a decrease in levels of brain-derived neutrophic factors (BDNF) [41]. Recently, estrogen signaling sensitivity has been proposed as a biomarker for PPD illustrating that the concentration of hormones may be less important than the mother’s sensitivity to and capacity for utilizing the hormones that are available [42].

Little is understood regarding the specific relationships between reproductive hormones and piglet-crushing events, and the research evaluating maternal hormones and piglet mortality are inconclusive [43,44]. What is known is that there is considerable individual variation in estrogen levels of sows surrounding parturition and estrogen levels have been reported to vary widely at the individual level, ranging from 3.5 to 9.0 ng/mL plasma [45]. Estradiol levels were observed to vary considerably between individual sows; higher estradiol levels postpartum were positively associated with aggressive behaviors, and were associated with lower progesterone levels on days leading up to farrowing [46].

Progesterone is theorized to be protective against depression because it has anxiolytic and anesthetic properties [47,48] and it also modulates the serotonergic receptors [49]. In a double-blind pregnancy-simulation study, synthetic estradiol and progesterone were administered to human women and then withdrawn, triggering symptoms of depression in women with a history of PPD, but not in women without a history of PPD [50]. No statistical differences were observed in the hormone levels between the two groups of women, but this study did highlight that women with a history of PPD may be more sensitive to the mood-destabilizing effects of changes in gonadal steroids.

Therefore, shifts in estrogen and progesterone during pregnancy and postpartum may contribute to PPD. In gilts, individuals who were extremely aggressive towards their piglets (e.g., attempted savaging) had higher estradiol to progesterone ratios postpartum and had lower progesterone levels on the days leading up to farrowing compared to sows that did not attempt to savage their piglets [46]. However, neither the rate nor the magnitude of the change in progesterone levels were observed to impact aggressive behavior towards piglets, so the combination of progesterone and estrogen may be more influential than their levels measured separately.

The magnitude of change in estrogen surrounding parturition has not been observed to impact the development of PPD [51,52], but higher estradiol and estriol levels postpartum had an inverse relationship with mother mood [53]. However, the diagnosis and etiology of PPD is variable, and thus women with very different symptomology may receive similar diagnoses [54,55]. Therefore, further understanding of the impact of the ratio between progesterone and estrogen surrounding parturition may enhance our understanding of PPD in humans and sows.

### 4.2. Oxytocin

Low levels of oxytocin during pregnancy or postpartum may be a risk factor for PPD. Human females with lower oxytocin levels during pregnancy [56] and at two and eight weeks postpartum [57] had more symptoms of postpartum psychosis. Lower oxytocin levels may predispose females to PPD, and could adversely impact nursing and lactation. In sows, oxytocin is associated with termination of nest building, initiation of farrowing, farrowing, lactation and nursing [58]. High levels of oxytocin have also been associated with increased levels of aggression in humans and animals. To evaluate this relationship in sows, blood samples and piglet mortality causes were collected from sows before and after farrowing. High risk sows (sows that had more pre-weaning mortality within the first 72 h post-farrowing) had higher average concentrations of oxytocin in the blood from samples taken daily using a catheter from 2d before through 2d post-farrowing [59], however, no differences were observed between high and low risk sows for concentrations of prolactin, cortisol, and urocortin. A positive relationship has been observed between oxytocin concentrations and sow unresponsiveness to piglet calls [60]. Therefore, if oxytocin dampens HPA reactivity during pregnancy, sows with higher oxytocin levels surrounding parturition may have a HPA that is responding abnormally to stress. However, several factors (e.g., litter size, piglet weight, farrowing duration and difficulty) may have contributed to the hormonal changes observed that are unrelated to the development of PPD in sows and further research is needed to elucidate these relationships.

In humans, mothers were more protective of their newborns after administration of intranasal oxytocin compared to a saline treatment during an invasive stranger test [61]. As sows may perceive animal handlers as a threat to their offspring, those with higher oxytocin levels may respond aggressively to their presence. Further, human mothers with PPD treated with intranasal oxytocin displayed less narcissism and had better acceptance of and interactions with the child [62]. Thus, oxytocin levels may also influence the sow’s perception of her new offspring and impact her behavior and acceptance toward them. Oxytocin levels that are either too high or too low may contribute to the development of PPD; however, more research is required to better understand this relationship.

### 4.3. Stress Hormones

Postpartum depression has been associated with abnormal HPA functioning, impaired cognitive abilities, and women with PPD are more likely to abuse their children and commit infanticide [39]. However, depression is multifaceted and can be caused by a combination of neurotransmitter disturbances, hormone dysregulations, genetics, and psychosocial factors [63,64]. Depressed patients show abnormal HPA axis function including a hyper secretion of cortisol, a flattened cortisol rhythm, prolonged recovery times after stress, and have lower morning and higher evening levels of cortisol [65,66]. Further, depressed patients have demonstrated abnormal responses to ACTH and CRH challenges, as well as atypical responses to dexamethasone suppression tests [67].

Several animal models have been utilized to demonstrate how depression impacts behavior. Depressed mice have demonstrated cognitive disturbances, sleep disruptions, anhedonia, weight disturbances, and increased immobility during a forced swim test [68]. Furthermore, mice experiencing a corticosterone-induced depressive state demonstrated changes in maternal care and increased depressive-like behaviors [69]. These mice with induced depressive states lost more body weight, spent more time off of the nest, spent less time nursing their pups, and had increased immobility in the forced swim test. Similar patterns of reduced maternal care can be observed in sows (e.g., restricting access to udder, more time lying ventrally, more restlessness, less piglet nosing, and more nursing bout terminations), suggesting that sows performing fewer maternal behaviors may be in a more depressive mental state.

Levels of corticotripin releasing hormones (CRH), adrenocorticotropic hormone (ACTH), and cortisol increase during pregnancy and then drop 4 days after delivery. CRH is secreted by the placenta, as CRH exponentially increases throughout gestation in a positive feedback loop. Women with accelerated CRH trajectories between 23 and 26 week had more PPD symptoms [70]. Larger drops in CRH from 36 week to 1 week postpartum as associated with less pronounced PPD symptoms [51]. Family support protected against PPD by dampening increases in CRH during pregnancy [71]. Therefore, identifying strategies that mitigate the activation of the HPA, or genetic selection for sows that minimizes the magnitude of stress hormone swing postpartum may provide benefit to humans and sows; yet further research is required to better elucidate these relationships.

### 4.4. Anxiety, Depression, Serotonin, and Genetics

Similar to other personality disorders, anxiety and depression appear to have a genetic component, suggesting certain individuals may be genetically predisposed to develop PPD. Many parallels have been observed between the development of PPD and anxiety disorders [72,73], and PPD appears to have a genetic component that impacts the serotonergic system, specifically the serotonin-transporter linked polymorphic region [74]. Women with PPD had lower affinity of platelet serotonin transporter binding sites [75]. Carriers of the long allele of the serotonin transporter gene 5-HTTLPR are more likely to receive a PPD diagnosis, and significant associations have been observed between variations of the 5-HTTLPR and COMPT genes (the genes that code for serotonin transporters and are responsible for maintaining appropriate levels of neurotransmitters in the prefrontal cortex) and PPD [76], suggesting that women with PPD have more difficulty transporting serotonin across cell membranes.

QTL mapping of maternal infanticide in sows showed that SSC2, SSC6, SSC14, SSC15, SSCX were associated with maternal infanticide and are known to regulate anxiety and bipolar disorder behaviors [77]. SSC2 is associated with the glucocorticoid receptor, and dysregulation of this region is associated with depression, bipolar disorder and schizophrenia. SSC6 is associated with serotonin and dopamine, susceptible to audio-genic epilepsies. SSC14 regulates neurotransmitter release from sympathetic nerves and from adrenergic neurons in the CNS associated with schizophrenia. SSCX regulates serotonin receptors [77]. Heritability estimates of piglet crushing have been reported to range from 0.03 to 0.06 from research and nucleus herds [78,79,80]. Farmer questionnaires reported sow carefulness around her piglets to range from 0.1 to 0.2. Therefore, there appears to be variation in maternal behavior, piglet-crushing prevalence, and the potential that this is a heritable trait that could be targeted for genetic selection.

Because behaviorally anxious sows spent more time ventral lying limiting accessibility to the udder that is important to piglets, and sows that crush offspring, also have increased restlessness, and are more aggressive towards piglets [81], relationships between behavior, genetics, and the serotonergic system may further elucidate the factors contributing to piglet crushing in sows.

## 5. Stressors

### 5.1. Gestational Stress

Gestational stress is a risk factor for developing PPD in multiple animal species. Mice administered exogenous corticosterone during gestation and postpartum experienced reduced litter sizes, reduced maternal care, and reduced cell proliferation in the dentate gyrus [13], an area of the hippocampus responsible for the formation of memories, and plays a large role in the development of psychosis and depression.

Gilts who experienced stress *in utero* also displayed aggressive behavior towards their own piglets post-farrowing. Stressed gilts also showed increased expression of corticotrophin releasing hormone in the paraventricular nucleus and amygdala [82], the areas of the brain responsible for mitigating anxiety and depression. Therefore, sows whose mother was stressed while they were fetuses demonstrated reduced maternal care.

Furthermore, stress during pregnancy can impact maternal behavior postpartum. Sows subjected to social stress (e.g., mixing) during mid-gestation took longer to respond to their piglet’s distress calls and spent more time lying ventrally, thus limiting the amount of time piglets could nurse [4]. Since stress in utero as well as stress during gestation can impact maternal behavior, identifying strategies to mitigate these stressful situations is paramount to good sow welfare.

### 5.2. Stress Due to Restrictions in Nestbuilding and Freedom of Movement

There is a mismatch between human and sow evolved adaptations and the altered features of their contemporary environments due to technological changes [83]; and these mismatches may be contributing to a growing number of medical and psychological problems. Women who decide to bottle feed over breastfeeding are at a higher risk of PPD; and routine hospital practices, including separating mother and baby immediately postpartum may be contributing to the prevalence of this disease [84,85]. This mismatch of evolutionary adaptation and biological motivation may cause stress that manifests itself in the performance of undesirable behaviors, including piglet crushing and savaging. 

During gestation, sows may be kept in either stalls or pens, contingent on the country in which they are housed. Regardless of the housing system, their movement is restricted. Prior to farrowing, sows are highly motivated to build a nest; and during the nest building process, the sow has been observed to collect nest building materials across a 50 m^2^ area surrounding the nest site [26].

For sows housed without the ability to move around or without access to nesting materials, their first experience with farrowing may be accompanied with undesirable side-effects, including altered maternal behaviors (e.g., piglet savaging, lower responsivity to piglet calls, shorter suckling periods, and psychological stress) [86]. Furthermore, the implementation of EU Directive 201/88/EC that requires sows to be loose-housed during the majority of gestation and allowing them to be subsequently transferred to farrowing crates may cause the sow to experience additional physiological and psychological stressors immediately prior to parturition due to the drastic restriction of space and mobility [86] during a time in which they are highly motivated to roam and nest build. While this may be an intuitive response when evaluating sow behavioral biology, research into the impact of farrowing systems on piglet mortality illustrate that piglet mortality is lower in farrowing crates compared to free farrowing systems [87] and in sows that are confined for four days post-farrowing [88]. Therefore, current housing and management practices need more evaluation regarding the impact of social environment, infrastructure features, and management decisions on pre-weaning piglet mortality.

## 6. Conclusions

As the swine industry selects for more socially tolerant sows to improve the welfare of group-housed sows, consideration should be made for the type of management implemented for these animals. If sows are highly social, they may need to be provided with a different social environment for farrowing and nursing than previously provided. Identifying management strategies that support gilts through parturition may reduce piglet-crushing rates and mitigate sow stress—which would be beneficial for producer productivity and overall sow welfare.

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
