# Peer review of "Parallels between Postpartum Disorders in Humans and Preweaning Piglet Mortality in Sows"

_animals, 2018, doi:10.3390/ani8020022_

Round 1

Reviewer 1 Report

The manuscript ‘Could piglet crushing in sows be used as a model for human post-partum depression’ provides a very interesting similarity between human post-partum depression and sow behavior and physiology during pregnancy, parturition and lactation. A similarity that indeed needs further investigation. From the review it is clear that sows in the swine industry may suffer from post-partum depression (PPD) too, and in this way humans could perhaps represent a model for sows as it seems we know already much about human PPD, but in my view the review does not make it really clear how or for what part of PPD sows could be a model for humans. The latter is not explained or highlighted much. Hence, I am not sure if the title suits this review very well. Furthermore, I have comments related to specific parts of the manuscript. The manuscript may certainly be suitable for publication in Animals, but the manuscript needs revision first. Below, I have listed my comments related to these specific parts: Line 11: postpartum, while in the title it is written as post-partum. Please be consistent in writing this word. Lines 57-61 and 63: ..... PP ..... should that not be ..... PPD .....? Lines 61-65: Please add references here. Line 65: I miss a brief outline of the review at the end of the introduction. Why do you review the factors explained in sections 2-5? Lines 67-77: You start here with humans, then go to animals and then to humans again. Perhaps this is deliberate, but I do not see the benefit then. Such a (quick) switch between subjects can also be found for instance at lines 115-127 or 173-178. The structure may be more clear and easier to read if you first discuss all things in humans and then in animals (or pigs in particular), or the other way around. Line 75: ... at 8wk postpartum ... But in line 49, PPD is defined as “any time during the pregnancy or the first four weeks postpartum”. I do not understand how PPD can develop at 8wk, while it is defined to only occur before 4wk postpartum. It is confusing and does not seem in line with each other. Please elucidate this. Line 98: Here you discuss postpartum psychosis, but earlier you used the term postpartum depression. It is not clear to me if in this case depression and psychosis mean the same illness and you can use both words interchangeably or that they refer to two different illnesses. Either way, I suggest to make it clear in the introduction how you regard this and how you refer to the illness. Lines 115-119: In humans you describe increases in estrogen products, while in sows estrogen levels decrease. This seems confusing with respect to the similarities between humans and sows you want to show, although I understand that the difference could be due to the fact that the time period of the hormone fluctuations is different (humans: pregnancy, sows: around parturition). However, if you want to convince the reader of the similarities between humans and sows, you might want to give a different example here? Line 145: had instead of has Lines 159-179: This part is a bit difficult to understand as both low and high levels of oxytocin seem to be a risk factor. Perhaps this part would benefit from adding a sentence at the end that more research is needed to understand the effects of oxytocin and thereby our understanding of PPD (or something like this), just as you did in lines 155-157. Line 178: I suggest to add ‘also’ after ‘may’ Lines 186 and 187: Looking up a normal cortisol rhythm (a peak in the morning which decreases during the day), it seems that a lower peak in the morning and higher levels in the evening come more or less down to a flattened rhythm. Therefore, are ‘a flattened cortisol rhythm’ and ‘lower morning and higher evening levels of cortisol‘ not redundant? Line 205: are instead of as Line 207: I miss a concluding sentence, such as in lines 155-157, here. Line 208: The link with hormones it not directly clear from this heading. I would suggest to revise the heading and include at least serotonin in it. Lines 211-212. The sympathetic system and HPA reactivity are not necessarily different between individuals with different coping styles, see for instance Koolhaas et al. (2010, Frontiers in Neuroendocrinology) and Coppens et al. (2010, Philosophical Transactions of the Royal Society B: Biological Sciences). As coping style is not further discussed in and does not seem important for the rest of the paragraph, and because in animals mainly only two coping styles are distinguished (proactive vs. reactive) which are not directly similar with the five major personality domains we distinguish in humans (neuroticism, extraversion, openness to experience, agreeableness, and conscientiousness), I would suggest to revise these lines by leaving coping styles out of the review. Line 218: I have no idea what COMPT genes are or play a role in. Could you add some information for this? Line 231: personality disorders instead of coping style, or alternatively, personality instead of coping style Line 234: PPD instead of PP Line 250: I suggest to change the heading into: Stress due to restrictions in nestbuilding and freedom of movement Line 253: The mentioning of ‘sunlight’ comes rather out of the blue and you also do not discuss it further. Is it needed to mention sunlight here? Line 264: Also here, I miss a concluding sentence.

Author Response

Reviewer 1 comments:

The manuscript ‘Could piglet crushing in sows be used as a model for human post-partum depression’ provides a very interesting similarity between human post-partum depression and sow behavior and physiology during pregnancy, parturition and lactation. A similarity that indeed needs further investigation. From the review it is clear that sows in the swine industry may suffer from post-partum depression (PPD) too, and in this way humans could perhaps represent a model for sows as it seems we know already much about human PPD, but in my view the review does not make it really clear how or for what part of PPD sows could be a model for humans.

The latter is not explained or highlighted much. Hence, I am not sure if the title suits this review very well.

Furthermore, I have comments related to specific parts of the manuscript. The manuscript may certainly be suitable for publication in Animals, but the manuscript needs revision first. Below, I have listed my comments related to these specific parts:

Line 11: postpartum, while in the title it is written as post-partum. Please be consistent in writing this word.

AU: the term “postpartum” has been used throughout the manuscript and appropriate edits have been made.

Lines 57-61 and 63: ..... PP ..... should that not be ..... PPD .....?

AU: corrected

Lines 61-65: Please add references here.

AU: corrected

Line 65: I miss a brief outline of the review at the end of the introduction. Why do you review the factors explained in sections 2-5?

AU: the following has been added to enhance clarity in New Lines 89-93: “. Examining the parallels in social network experiences, parity contributions, hormone shifts, and behavioral changes or restrictions between PPD in humans and piglet crushing in sows presents an opportunity to enhance our understanding of both issues, use lessons learned from humans to mitigate piglet crushing in sows, and perhaps increase our understanding of human PPD based upon what we can learn from sows.”

Lines 67-77: You start here with humans, then go to animals and then to humans again. Perhaps this is deliberate, but I do not see the benefit then. Such a (quick) switch between subjects can also be found for instance at lines 115-127 or 173-178. The structure may be more clear and easier to read if you first discuss all things in humans and then in animals (or pigs in particular), or the other way around.

AU: I can see how there could be a personal preference for this style of writing, however, the switch between humans and animals within a single section was intentional.  The author felt as though the manuscript would read well by addressing a single issue from both the human and pig side, rather than presenting the same issues twice, once from a human perspective and the other from the animal’s. 

Line 75: ... at 8wk postpartum ... But in line 49, PPD is defined as “any time during the pregnancy or the first four weeks postpartum”. I do not understand how PPD can develop at 8wk, while it is defined to only occur before 4wk postpartum. It is confusing and does not seem in line with each other. Please elucidate this.

AU:  The author has revised the sentence to enhance clarity.  New lines 100-103: “Social support, such as perceived assistance from a partner and emotional closeness with other mothers in the first few weeks after parturition were observed to be key factors in predicting the development of PPD in mothers that were evaluated 8wk postpartum [24].”

Line 98: Here you discuss postpartum psychosis, but earlier you used the term postpartum depression. It is not clear to me if in this case depression and psychosis mean the same illness and you can use both words interchangeably or that they refer to two different illnesses. Either way, I suggest to make it clear in the introduction how you regard this and how you refer to the illness.

AU: at the beginning of the manuscript, I define PPD as Postpartum disorder.  PPD includes postpartum depression and postpartum psychosis.  Therefore, to enhance clarity, the first sentence of this paragraph has been revised.

Lines 115-119: In humans you describe increases in estrogen products, while in sows estrogen levels decrease. This seems confusing with respect to the similarities between humans and sows you want to show, although I understand that the difference could be due to the fact that the time period of the hormone fluctuations is different (humans: pregnancy, sows: around parturition). However, if you want to convince the reader of the similarities between humans and sows, you might want to give a different example here?

AU:  I believe there may be some confusion regarding what was presented.  I illustrate that that estrogen products increase drastically during pregnancy in humans and then talk about how estrogen drops off after farrowing in sows.  Therefore, I am not making the claim that they make opposite swings.  This is a representation of parallel hormone shifts.  A sentence has been added to the end of the paragraph to further clarify this point.  New lines 138-140: “Both humans and sows have higher levels of estrogen products during gestation, and these hormone levels will drop drastically postpartum.  Therefore, the impact of these hormonal changes may be similar in humans and sows.”

Line 145: had instead of has

AU:  corrected

Lines 159-179: This part is a bit difficult to understand as both low and high levels of oxytocin seem to be a risk factor. Perhaps this part would benefit from adding a sentence at the end that more research is needed to understand the effects of oxytocin and thereby our understanding of PPD (or something like this), just as you did in lines 155-157.

AU:  New Lines 270-271: “Oxytocin levels that are either too high or too low may contribute to the development of PPD; however, more research is required to better understand this relationship.”

Line 178: I suggest to add ‘also’ after ‘may’

AU:  corrected

Lines 186 and 187: Looking up a normal cortisol rhythm (a peak in the morning which decreases during the day), it seems that a lower peak in the morning and higher levels in the evening come more or less down to a flattened rhythm. Therefore, are ‘a flattened cortisol rhythm’ and ‘lower morning and higher evening levels of cortisol‘ not redundant?

AU:  These are describing different phenomena.  The lower peak in morning and high in evening is an inverse HPA functioning, while a flattened cortisol rhythm describes HPA functioning that does not fluctuate throughout the day.  A normally functioning HPA system will have higher in the morning and lower in the evening, however, those with inverse or non-functioning are possible.

Line 205: are instead of as

AU:  corrected

Line 207: I miss a concluding sentence, such as in lines 155-157, here.

AU: A concluding sentence has been added to New Lines 297-299: “Therefore, identifying strategies that mitigate the activation of the HPA, or genetic selection for sows that minimizes the magnitude of stress hormone swing postpartum may provide benefit to humans and sows; yet further research is required to better elucidate these relationships.”

Line 208: The link with hormones it not directly clear from this heading. I would suggest to revise the heading and include at least serotonin in it.

AU: this has been revised to “Anxiety, depression, serotonin, and genetics”

Lines 211-212. The sympathetic system and HPA reactivity are not necessarily different between individuals with different coping styles, see for instance Koolhaas et al. (2010, Frontiers in Neuroendocrinology) and Coppens et al. (2010, Philosophical Transactions of the Royal Society B: Biological Sciences). As coping style is not further discussed in and does not seem important for the rest of the paragraph, and because in animals mainly only two coping styles are distinguished (proactive vs. reactive) which are not directly similar with the five major personality domains we distinguish in humans (neuroticism, extraversion, openness to experience, agreeableness, and conscientiousness), I would suggest to revise these lines by leaving coping styles out of the review.

AU: Thank you for the thoughtful comments.  Per your recommendation, I have deleted references to coping style from this section.

Line 218: I have no idea what COMPT genes are or play a role in. Could you add some information for this?

AU: New lines 308-310: “…observed between variations of the 5-HTTLPR and COMPT genes (the genes that code for serotonin transporters and are responsible for maintaining appropriate levels of neurotransmitters in the prefrontal cortex) and PPD…”

Line 231: personality disorders instead of coping style, or alternatively, personality instead of coping style

AU: Personality has been added to replace coping style

Line 234: PPD instead of PP

AU:  corrected

Line 250: I suggest to change the heading into: Stress due to restrictions in nestbuilding and freedom of movement

AU:  corrected

Line 253: The mentioning of ‘sunlight’ comes rather out of the blue and you also do not discuss it further. Is it needed to mention sunlight here?

Au: this portion of the sentence has been deleted.  The author was referencing the need for vitamin D as a preventative measure of PPD in humans, but it is unknown how vitamin D impacts sows.

Line 264: Also here, I miss a concluding sentence. 

AU:  corrected

Reviewer 2 Report

The paper entitled “ Could piglet crushing in sows be used as a model for human post-partum depression?” is well-written and of interest for readers.

The major concern regarding the paper is maybe clarifying a bit more the objectives of the review paper. The title suggests that the main objective is to present piglet crushing as a model of PPD in humans, whereas the summary of the paper is on changes in pig industry that could help reduce piglet crushing (based on knowledge gained from PPD in humans). Maybe the title and objectives should clarify more that the paper is on the comparison between the similar patterns seen in both situations.

Few specific comments:

Lines 59 to 65. PP is PPD in some of the sentences??

Line 104. Sows refers to primiparous sows (gilts)?

Line 104-112. Although the authors mention it before, caution has to be taken because sows certainly seek for isolation during a few days around parturition. So it has to be clear that this social support refers to pre-partum and post-partum when sows join the group again (around 10 days after delivery)

Line 146. Had instead of has

Author Response

Reviewer 2

The paper entitled “ Could piglet crushing in sows be used as a model for human post-partum depression?” is well-written and of interest for readers.

The major concern regarding the paper is maybe clarifying a bit more the objectives of the review paper. The title suggests that the main objective is to present piglet crushing as a model of PPD in humans, whereas the summary of the paper is on changes in pig industry that could help reduce piglet crushing (based on knowledge gained from PPD in humans). Maybe the title and objectives should clarify more that the paper is on the comparison between the similar patterns seen in both situations.

AU: The title has been revised, and objectives clarified.

Few specific comments:

Lines 59 to 65. PP is PPD in some of the sentences??

AU: corrected

Line 104. Sows refers to primiparous sows (gilts)?

AU:  corrected

Line 104-112. Although the authors mention it before, caution has to be taken because sows certainly seek for isolation during a few days around parturition. So it has to be clear that this social support refers to pre-partum and post-partum when sows join the group again (around 10 days after delivery)

AU:  Thank you for this observation.  Additions have been added to New Lines 105-138 to address this comment.

Line 146. Had instead of has

AU:  corrected

Reviewer 3 Report

General comments

The author opens a very interesting and new issue. Despite of this, the author forgets to discuss in each section the main issue of the text (crushing events). For instance, from L94 to L228 there isn’t any mention of crushing. Is that because there isn’t any evidence or previous studies on that? In that case, I appreciate to said it.

From your manuscript, few evidences confirm the relationship between crushing events in sows and PPD, but that is not discussed in each section. I propose to focus your paper connecting savaging with PPD and open the possibility that crushing events also could be related with PPD.

The title is to ambitious, or at least, the author doesn’t answer it, neither explain which are the benefits to use sow model for PPD. Is that viable? Which are the advantages and disadvantages to propose this model? Which are the next steps to be done in research?

Major changes

Social Network (L78-88): It is necessary to explain that sows in natural conditions (and similar to the wild boar) leave the flock at least 24h before parturition. They look for a right place to farrow and construct the nest. Afterwards, around 9-10 days after parturition, the sow return to the flock. (example of reference: Jensen 1986; Observations on the maternal behaviour of free-ranging domestic pigs). Maybe this return to the flock 9-10 days after parturition is comparable with the described importance of the “emotional closeness with other mothers in humans after labour” The anecdotal observations of group housed sows (L83-L87) should be deleted or confirm with extra-bibliography. The management and type of facilities (density, distribution of eating-resting areas…) could enhance the social interaction around parturition.

Impact of parity: Please, also specify which are the factors affecting primiparous women to be more in risk to develop postpartum psychosis. Are those factors similar to the described factors for sows?

Reproductive hormones: Few evidences are shown connecting hormone profile and crushing events. Introduce specific references, or said that there isn’t any evidence in sows.

Please, be more precise when explain results related to the Phillips et al. (2014) study. For instance, in L167-169, higher concentrations of oxytocin are probably related to a higher duration of parturition, and consequently a higher number of weak piglets born (higher number of risky piglets to be crush). In addition, no differences were found comparing risky sows vs. non-risky sows in other hormones (cortisol, prolactin…)

Stressors: In the nestbuilding and freedom of movement section, please, compare crushing events between sows allocated in crates and sows allocated in free-farrowing systems. Results may be contradictory with your hypothesis.

Summary: There isn’t any connection between the summary and your paper (and the title of the manuscript)!

Minor changes

Please, check the Instructions for Authors. For example, references must be numbered in the text.

Abbreviations (PP and PPD) should be defined not only in the abstract, but also in the main text.

L55-L59: phrase too long

L59-L65: check that you want to talk about PP instead of PPD. For instance, L59 “predictors of PP in humans….” Do you mean, “predictors of PPD I humans”?

L79 and elsewhere: limit the use of “labour” for humans. Use “parturition” or “farrowing/farrow” for sows

Is there any difference between psychosis and depression? The author seems to use these words as synonymous. 

Author Response

Reviewer 3

The author opens a very interesting and new issue. Despite of this, the author forgets to discuss in each section the main issue of the text (crushing events). For instance, from L94 to L228 there isn’t any mention of crushing. Is that because there isn’t any evidence or previous studies on that? In that case, I appreciate to said it.

AU: I am confused by this comment, as piglet crushing is referenced in OLD lines 99-102, 144-150, 164-170, 221-227.

From your manuscript, few evidences confirm the relationship between crushing events in sows and PPD, but that is not discussed in each section. I propose to focus your paper connecting savaging with PPD and open the possibility that crushing events also could be related with PPD.

AU: while I believe this comment is valid, I believe that the challenges associated with sows not responding to their own piglet screaming is a behavioral manifestation of apathy.  Piglet death post-farrowing may be due to active aggressive acts towards the piglets (savaging), or apathy and neglect towards them (crushing).  Both could be considered emotional affective disorders that may just differ in the severity of their manifestation or individual differences in their phenotypic performance.

The title is to ambitious, or at least, the author doesn’t answer it, neither explain which are the benefits to use sow model for PPD. Is that viable? Which are the advantages and disadvantages to propose this model? Which are the next steps to be done in research?

AU: The title has been revised to better represent the content of the text.

Major changes

Social Network (L78-88): It is necessary to explain that sows in natural conditions (and similar to the wild boar) leave the flock at least 24h before parturition. They look for a right place to farrow and construct the nest. Afterwards, around 9-10 days after parturition, the sow return to the flock. (example of reference: Jensen 1986; Observations on the maternal behaviour of free-ranging domestic pigs). Maybe this return to the flock 9-10 days after parturition is comparable with the described importance of the “emotional closeness with other mothers in humans after labour” The anecdotal observations of group housed sows (L83-L87) should be deleted or confirm with extra-bibliography. The management and type of facilities (density, distribution of eating-resting areas…) could enhance the social interaction around parturition.

AU: Thank you for the constructive criticism.  The paragraph has been revised to address those concerns above and clarify the point of the paragraph.  New lines 105-138: “A similar argument can be made for pigs.  Pigs are a social, highly cognitive species, and with the evolution of social structures comes the capacity to communicate with conspecifics and transfer social information either by experience or observation.  Pigs will separate themselves from the group close to parturition, they may begin parturition well before delivery.  Even though pigs will seek isolation the few days surrounding parturition, the social support prior to and after the sow rejoins the group may be invaluable to the success of the sow and her litter [24].  Free ranging pigs have been observed to organize themselves into small social clusters, or sounders, that usually are comprised of a number of adult sows with their daughters, and any unweaned young [25].  The nature of this social organization structure depicts a scenario in which young breeding-age females learn appropriate maternal behaviors from their mothers.  Sows in natural conditions will leave the sounder at least 24h prior to farrowing and will begin to construct a nest. Approximately 9-10 days post-farrowing, the sow will return to their sounder with the newborn piglets [26].  Group housed sows displayed improved maternal behavior when housed in lactation groups 3 days post-farrowing compared to sows housed in farrowing crates [24].  The development of housing and management technologies that allow for social contact surrounding farrowing [27] further emphasizes the need to better understand the social dynamics surrounding farrowing on sow and piglet welfare.  While more research is needed evidence suggests that the social environment, physical contact opportunities, and information transfer may be valuable to the success and mitigation of PPD in the primiparous sow.”

Impact of parity: Please, also specify which are the factors affecting primiparous women to be more in risk to develop postpartum psychosis. Are those factors similar to the described factors for sows?

AU:  Little is known regarding the full impact of parity on both sows and humans.  However, there are some similarities and differences.  I made an attempt to clarify both of these points in New Lines 147-165: “Both primiparous sows and humans are at an increased risk of developing PPD.  Primiparous humans are more likely to develop puerpareal psychosis [30, 31], yet, the development of PPD in humans is multifactorial and likely epigenetic in development [32].  Primiparous sows have been observed to savage more [33], and be more likely to crush their piglets compared to second and third-parity sows [34].  Human mothers with prior experience caring for infants are less likely to develop PPD [35], and rates of PPD in sows are lower in multiparous parity sows.  However, consideration should be made to the impact of management practices because sows that crush a large number of piglets may be culled after a single parity.

For sows, the factors contributing to this phenomenon are multifactorial and are potentially exacerbated by the fact that primiparous sows (gilts) in commercial production systems may not have exposure to multiparous sows surrounding parturition, and may therefore, have no information about what to expect. Primiparous sows may not have seen a piglet since they were a piglet, so the sudden appearance of a piglet after experiencing farrowing may be very stressful.  The stressors associated with the novelty of parturition and movement from a familiar gestation environment to a novel farrowing environment, the discomfort associated with pain, the lack of understanding why they are experiencing the pain associated with farrowing, the impact of large hormonal swings, and the lack of social support in a highly social species may contribute to the development of a poor affective state.  Therefore, lack of experience engaging with and caring for young may be a contributing factor to the manifestation of PPD in humans and sows.”

Reproductive hormones: Few evidences are shown connecting hormone profile and crushing events. Introduce specific references, or said that there isn’t any evidence in sows.

AU: New lines 205-207: “Little is understood regarding the specific relationships between reproductive hormones and piglet crushing events, and the research evaluating maternal hormones and piglet mortality are inconclusive [44, 45].  What is known is that there is considerable …”

Please, be more precise when explain results related to the Phillips et al. (2014) study. For instance, in L167-169, higher concentrations of oxytocin are probably related to a higher duration of parturition, and consequently a higher number of weak piglets born (higher number of risky piglets to be crush). In addition, no differences were found comparing risky sows vs. non-risky sows in other hormones (cortisol, prolactin…)

AU: New lines 239-252: “In sows, oxytocin is associated with termination of nest building, initiation of farrowing, farrowing, lactation and nursing [59].  High levels of oxytocin have also been associated with increased levels of aggression in humans and animals.  To evaluate this relationship in sows, blood samples and piglet mortality causes were collected from sows before and after farrowing.  High risk sows (sows that had more pre-weaning mortality within the first 72 hours post-farrowing) had higher average concentrations of oxytocin in the blood from samples taken daily using a catheter from 2d before through 2d post-farrowing [60], however, no differences were observed between high and low risk sows for concentrations of prolactin, cortisol, and urocortin.  A positive relationship has been observed between oxytocin concentrations and sow unresponsiveness to piglet calls [61]. Therefore, if oxytocin dampens HPA reactivity during pregnancy, sows with higher oxytocin levels surrounding parturition may have a HPA that is responding abnormally to stress.  However, several factors (e.g., litter size, piglet weight, farrowing duration and difficulty) may have contributed to the hormonal changes observed that are unrelated to the development of PPD in sows and further research is needed to elucidate these relationships.”

Stressors: In the nestbuilding and freedom of movement section, please, compare crushing events between sows allocated in crates and sows allocated in free-farrowing systems. Results may be contradictory with your hypothesis.

AU: New lines: 386-391:” While this may be an intuitive response when evaluating sow behavioral biology, research into the impact of farrowing systems on piglet mortality illustrate that piglet mortality is lower in farrowing crates compared to free farrowing systems [85] and in sows that are confined for four days post-farrowing [86].  Therefore, current housing and management practices need more evaluation regarding the impact of social environment, infrastructure features, and management decisions on pre-weaning piglet mortality.”

Summary: There isn’t any connection between the summary and your paper (and the title of the manuscript)!

AU: The summary and title have been revised to better reflect the content of the paper.

Minor changes

Please, check the Instructions for Authors. For example, references must be numbered in the text.

AU: corrected

Abbreviations (PP and PPD) should be defined not only in the abstract, but also in the main text.

AU: corrected

L55-L59: phrase too long

AU: New lines 79-83:” Maternal infanticide is a counter-intuitive, counter-evolutionary behavior accompanied by serious welfare and economic ramifications.   Hagen [13] proposed that PPD facilitated maternal disinvestment in offspring that are unlikely to survive and later reproduce, and that the development of PPD broadcasts the mother’s need for support.”

L59-L65: check that you want to talk about PP instead of PPD. For instance, L59 “predictors of PP in humans….” Do you mean, “predictors of PPD I humans”?

AU: corrected

L79 and elsewhere: limit the use of “labour” for humans. Use “parturition” or “farrowing/farrow” for sows

AU: corrected

Is there any difference between psychosis and depression? The author seems to use these words as synonymous

AU: this has been addressed at the beginning of the manuscript in New Lines 33-34: “some mothers may develop postpartum disorders (PPD), including postpartum depression and postpartum psychosis.”

Reviewer 4 Report

See attached.

Author Response

Reviewer 4

Review for Animals

“Could piglet crushing in sows be used as a model for human post-partum depression?”

This is an exceptionally well written paper that describes some intriguing similarities between postpartum depression in humans and different aspects of piglet crushing by sows.

Emerging evidence shows that mismatches between evolved adaptations and altered features of the contemporary environment due to technological changes may be the source of a growing number of medical and psychological problems. I think the present paper could be strengthened further by incorporating the mismatch hypothesis of postpartum depression in humans (Gallup, Pipitone, Carrone, & Leadholm, 2010). The decision to substitute bottle feeding for breastfeeding puts many women out of phase with an important aspect of their evolutionary history. Bottle feeding also appears to unwittingly simulate child loss, and there is growing evidence that bottle feeding is a significant risk factor for post-partum depression in women as well as a variety of other neonatal and maternal problems (Gallup, Spaulding, & Aboul-Seoud, 2016 ). Routine hospital practices such as separating mothers from their babies by keeping babies in nurseries may also simulate child loss and contribute to postpartum depression.

Domestication and captivity may also entail many mismatches. The last section of the manuscript on nest building and freedom of movement could greatly benefit from an extended discussion of the mismatch hypothesis as it applies to a variety of similar aspects of modern animal husbandry (e.g., restrictions on social interaction, roaming, nest building, lose housing during gestation) that may be related to postpartum depression in sows.

AU:  Thank you for the helpful feedback.  The mismatch hypothesis has been further elaborated upon in New Lines 357-364: “There is a mismatch between human and sow evolved adaptations and the altered features of their contemporary environments due to technological changes [84]; and these mismatches may be contributing to a growing number of medical and psychological problems.  Women who decide to bottle feed over breastfeeding are at a higher risk of PPD; and routine hospital practices, including separating mother and baby immediately postpartum may be contributing to the prevalence of this disease [85, 86].  This mismatch of evolutionary adaptation and biological motivation may cause stress that manifests itself in the performance of undesirable behaviors, including piglet crushing and savaging.”

Round 2

Reviewer 3 Report

The author addressed all suggestion given.

The manuscript is recommended to publish after few changes:

Abstract:

L21 define PPD. Write: postpartum depression (PPD)

L23. You can directly use PPD, without postpartum depression

Introduction and elsewhere:

L38. Introduce that primiparous sows and gilts are synonymous. Write: primiparous sows (gilts)

Use primiparous sows or gilts in the rest of the manuscript. Both concepts are not used together (for example delete L100 primiparous gilts and use primparous sows or gilts)

Summary:

I think the summary don’t match with the text. Here, summary is focused to the swine industry problems without talking about the possible parallelism with disorders in women. The author has different possibilities to address it:

(1)   Extend the summary introducing the parallelism with disorders in women

(2)   Delete summary (I think is possible, according to the guidelines of the journal), as your “simple summary” perfectly addressed the issue

(3)   Replace “summary” with “conclusions”, which could be more useful in your review.